# Inverse decision-making using neural amortized Bayesian actors

**Dominik Straub**[*,1,2]**, Tobias F. Niehues**[*,1,2]**, Jan Peters**[2,3] **& Constantin A. Rothkopf**[1,2,3]

[1] Institute of Psychology, Technical University of Darmstadt

[2] Centre for Cognitive Science, Technical University of Darmstadt

[3] Department of Computer Science, Technical University of Darmstadt & Hessian Center for Artificial Intelligence

`{firstname.lastname}@tu-darmstadt.de`

## Abstract

Bayesian observer and actor models have provided normative explanations for many behavioral phenomena in perception, sensorimotor control, and other areas of cognitive science and neuroscience. They attribute behavioral variability and biases to interpretable entities such as perceptual and motor uncertainty, prior beliefs, and behavioral costs. However, when extending these models to more naturalistic tasks with continuous actions, solving the Bayesian decision-making problem is often analytically intractable. Inverse decision-making, i.e. performing inference over the parameters of such models given behavioral data, is computationally even more difficult. Therefore, researchers typically constrain their models to easily tractable components, such as Gaussian distributions or quadratic cost functions, or resort to numerical approximations. To overcome these limitations, we amortize the Bayesian actor using a neural network trained on a wide range of parameter settings in an unsupervised fashion. Using the pre-trained neural network enables performing efficient gradient-based Bayesian inference of the Bayesian actor model's parameters. We show on synthetic data that the inferred posterior distributions are in close alignment with those obtained using analytical solutions where they exist. Where no analytical solution is available, we recover posterior distributions close to the ground truth. We then show how our method allows for principled model comparison and how it can be used to disentangle factors that may lead to unidentifiabilities between priors and costs. Finally, we apply our method to empirical data from three sensorimotor tasks and compare model fits with different cost functions to show that it can explain individuals' behavioral patterns.

## 1 Introduction

Explanations of human behavior based on Bayesian observer and actor models have been widely successful, because they structure the factors influencing behavior into interpretable components (Ma, 2019). They have explained a wide range of phenomena in perception (Weiss et al., 2002; Ernst & Banks, 2002; Wei & Stocker, 2015; Kersten et al., 2004), motor control (Körding & Wolpert, 2004; Todorov & Jordan, 2002), other domains of cognitive science (Griffiths & Tenenbaum, 2006; Xu & Tenenbaum, 2007), and neuroscience (Behrens et al., 2007; Berkes et al., 2011). Bayesian observer models assume that an actor receives uncertain sensory information about the world. This sensory information is uncertain because of ambiguity of the sensory input or noise in neural responses (Kersten et al., 2004). To obtain a belief about the state of the world, the actor fuses this information with prior knowledge according to Bayes' rule. But humans do not only form beliefs about the world, they also act in it. In Bayesian actor models, this is expressed as the minimization of a cost function, which expresses the actor's goals and constraints. An optimal actor should minimize this cost function while taking their belief about the state of the world into account. However, there is

---

[*]Both authors contributed equally to this paper.

not only uncertainty in perception but also in action outcomes, e.g. due to the inherent variability of the motor system (Van Beers et al., 2004). If an actor wants to perform the best action possible, this uncertainty should also be incorporated into the decision-making process by integrating over the distribution of action outcomes (Trommershäuser et al., 2008).

One problem, which makes solving the Bayesian decision-making problem hard in practice, is that the expected cost is often not analytically tractable, because it involves integrals over the posterior distribution and the action distribution. Consequently, optimization of the expected cost is also often intractable. Certain special cases, especially Gaussian distributions and quadratic cost functions, admit analytical solutions. Applications of Bayesian models have typically made use of these assumptions. However, empirical evidence shows that the human sensorimotor system does not conform to these assumptions. Noise in the motor system depends on the force produced (Harris & Wolpert, 1998; Todorov & Jordan, 2002) and variability in the sensory system follows a similar signal-dependence known as Weber's law (Weber, 1831). Cost functions other than quadratic costs have been shown to be required in sensorimotor tasks (Körding & Wolpert, 2004; Sims, 2015). If we want to model what is known about the human sensorimotor system or treat more naturalistic task settings, we need to incorporate these non-Gaussian distributions and non-quadratic costs.

Another challenge is that Bayesian actor models often have free parameters. One option is to set these parameters by hand, e.g. often assuming that subjects use the identical prior and cost function defined by the experiment, as in ideal observer analysis, or by matching prior distributions to the statistics of the natural environment. Predictions of the Bayesian model are then compared to behavioral data to assess optimality. This practice is problematic, because priors and costs may not be known to the experimenter beforehand and can be idiosyncratic to individual participants. For example, an actor's cost function might not only contain task goals, e.g. hitting a target, but also internal costs. These can include cognitive factors such as computational resources (Lewis et al., 2014; Lieder & Griffiths, 2020; Gershman et al., 2015), but also more generally cognitive and physiological factors including biomechanical effort (Hoppe & Rothkopf, 2016; Straub & Rothkopf, 2022). An actor's prior distribution might neither match the statistics of the task at hand nor those of the natural environment (Feldman, 2013). In the spirit of rational analysis (Simon, 1955; Anderson, 1991; Gershman et al., 2015), this has motivated researchers to invert Bayesian actor models, i.e. to use Bayesian actor models as statistical models of behavior and perform Bayesian inference over the free parameters from behavior. This approach has come to be known in different application areas under different names, including doubly-Bayesian analysis (Aitchison et al., 2015), cognitive tomography (Houlsby et al., 2013), inverse reinforcement learning (Rothkopf & Ballard, 2013; Muelling et al., 2014) and inverse optimal control (Kwon et al., 2020; Schultheis et al., 2021). However, because the forward problem of computing optimal Bayesian actions for a given perception and decision-making problem is computationally expensive, the inverse decision-making problem, if only numerical solutions are available, is prohibitively expensive and makes computing gradients with respect to the parameters for efficient optimization or sampling infeasible.

Here, we address these issues by providing a new method for inverse decision-making in sensorimotor tasks with continuous actions. Such tasks are widespread in cognitive science, psychology, and neuroscience and include so-called production, reproduction, magnitude estimation, and adjustment tasks. First, we formalize such tasks with Bayesian networks, both from the perspective of the researcher and from the perspective of the participant. Second, we approximate the solution of the Bayesian decision-making problem with a neural network, which is trained in an unsupervised fashion using the decision problem's cost function as a stochastic training objective. Third, using the pre-trained neural network as a stand-in for the Bayesian actor within a statistical model enables efficient Bayesian inference of the Bayesian actor model's parameters given observed behavior. Fourth, we show on simulated datasets that the posterior distributions obtained using the neural network recover the ground truth parameters very closely to those obtained using the analytical solution for various typical response patterns like undershoots or regression to the mean behavior. Fifth, we show how posterior distributions over the actor's internal parameters can be used to perform principled model comparison. Sixth, identifiability problems between priors and costs of Bayesian actor models are investigated, which can now be resolved based on our proposed method. Finally, we apply our method to human behavioral data from three different experiments and show that the inferred cost functions explain the previously mentioned typical behavioral patterns not only in synthetically generated but also empirically observed data.

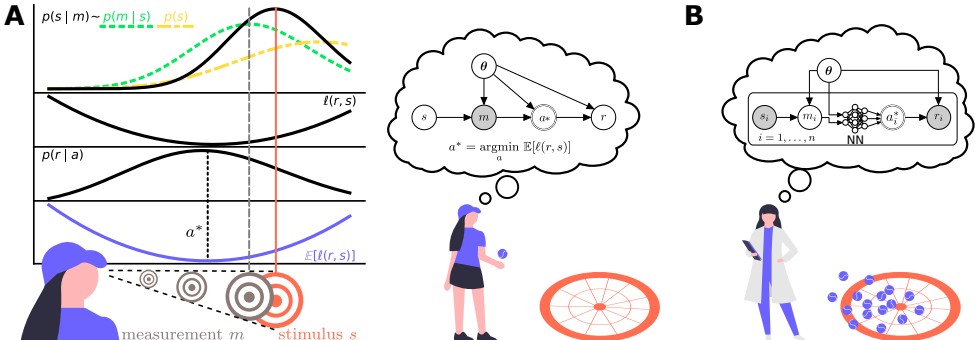

Figure 1: **A** Bayesian decision-making problem from the perspective of the subject. The subject needs to find the optimal action $a^*$ based on the sensory measurement $m$ of the state of the world $s$. Combined with a cost function $\ell(r, s)$ and the action distribution $r \sim \text{Lognormal}(a, \sigma_r)$, they want to minimize a cost function under their belief about the state of the world which yields the optimal action $a^* = \arg\min_a \mathbb{E}_{p(s \mid m)} \left[ \mathbb{E}_{p(r \mid a)} \left[ \ell(r, s) \right] \right]$. **B** Bayesian inference problem about the subject's parameters from the perspective of the researcher. The researcher solves the inverse decision-making problem, i.e. they want to infer the posterior distribution $p(\boldsymbol{\theta} \mid \mathcal{D})$ over the parameters $\boldsymbol{\theta}$ of the subject's perception-action system and cost function given a dataset $\mathcal{D} = \{s_i, r_i : i = 1, \ldots, n\}$ of stimuli $s_i$ and responses $r_i$ from $n$ trials. To make inference of the posterior over $\boldsymbol{\theta}$ feasible, we use a neural network as an approximator for the optimal action $a^*$.

## RELATED WORK

Inferring priors and costs from behavior has been a problem of interest in cognitive science for many decades. In psychophysics, for example, signal detection theory is an early example of an application of a Bayesian observer model used to estimate sensory uncertainty and a criterion, which encompasses prior beliefs and a particular cost function (Green et al., 1966). Psychologists and behavioral economists have developed methods to measure the subjective utility function from economic decisions (Tversky & Kahneman, 1992). More recently, Bayesian actor models have been used within statistical models to infer parameters of the observer's likelihood function (Girshick et al., 2011), the prior (Stocker & Simoncelli, 2006; Girshick et al., 2011; Sohn & Jazayeri, 2021), and the cost function (Körding & Wolpert, 2004; Sims, 2015; Sohn & Jazayeri, 2021). The inference methods are often bespoke tools for the specific model considered in a study. They are also typically limited to discrete decisions and cannot be applied to continuous actions.

There are two notable exceptions. Acerbi et al. (2014) presented an inference framework for Bayesian observer models using mixtures of Gaussians. While their approach only takes perceptual uncertainty into account in the decision-making process, we assume that the agent also considers action variability here. Furthermore, instead of only inverted Gaussian mixture cost functions, our method allows for arbitrary, parametric cost functions that are easily interpretable. Neupärtl & Rothkopf (2021) introduced the idea of approximating Bayesian decision-making with neural networks. They trained neural networks in a supervised fashion using a dataset of numerically optimized actions. We extend this approach in three ways. First, we train the neural networks directly on the cost function of the Bayesian decision-making problem without supervision, overcoming the necessity for computationally expensive numerical solutions. Second, in addition to cost function parameters and motor variability, we also infer priors and sensory uncertainty. Finally, we leverage the differentiability of neural networks in order to apply efficient gradient-based Bayesian inference methods, allowing us to draw thousands of samples from the posterior over parameters in a few seconds.

Our method is also related to amortized and likelihood-free inference (Fengler et al., 2021; Greenberg et al., 2019; Govindarajan et al., 2022; Radev et al., 2020). A conceptual difference is that we amortize the solution of the Bayesian decision-making problem faced by a subject instead of the inference process itself. This allows us to solve the Bayesian inference problem from the perspective of a researcher using efficient gradient-based inference techniques, without the need to use amortized or likelihood-free inference.

## 2 BACKGROUND: BAYESIAN DECISION-MAKING

We start with the standard formulation of Bayesian decision theory (Berger, 1985), illustrated in Fig. 1 A. An actor receives a stochastic observation $m$ generated from a latent state $s$, according to a generative model $p(m \mid s)$. Since the actor has no direct access to the true value of the state $s$, they need to infer it by combining a prior distribution $p(s)$ with the likelihood $p(m \mid s)$ using Bayes' rule $p(s \mid m) \propto p(m \mid s) \, p(s)$. This Bayesian inference process describes the actor's perception. Based on their perception, the actor's goal is to perform an optimal action $a^*$, which represents the intended motor response. This is commonly framed as a decision-making problem based on a cost function $\ell(a, s)$. The optimal action $a^*$ for the actor is the action $a$ that minimizes the expected cost under the posterior distribution

$$a^* = \arg\min_a \int \ell(a, s) \, p(s \mid m) \, ds. \tag{1}$$

Often, the loss is not defined directly in terms of the performed action $a$, but in terms of the expectation over some stochastic version of it $r \sim p(r \mid a)$, modeling variability in responses with the same intended action $a$. Taking the expectation over the posterior distribution and the response distribution lets us expand Eq. (1) as

$$a^* = \arg\min_a \int \int \ell(r, s) \, p(r \mid a) \, p(s \mid m) \, dr \, ds, \tag{2}$$

which changes the problem conceptually from computing an estimate of a latent variable to performing an action that is subject to motor noise.

## 3 METHOD

The Bayesian decision-making problem in Section 2 describes the situation faced by a subject performing a task and the optimal solution to it. From the researcher's perspective, we now want to infer the parameters of the subject's perception-action system. For example, we might be interested in the subject's perceptual uncertainty and their prior belief about possible target location, their action variability, and the cost of extending effort for motor responses. These parameters constitute a parameter vector $\boldsymbol{\theta}$. Formally, we want to compute the posterior $p(\boldsymbol{\theta} \mid \mathcal{D})$ for a set $\mathcal{D}$ of behavioral data, as shown in Fig. 1 B.

Our proposed method consists of two parts. First, we approximate the optimal solution of the Bayesian decision-making problem with a neural network (Section 3.1). A forward-pass of the neural network is very fast compared to the computation of numerical solutions to the original Bayesian decision-making problem, and gradients of the optimal action w.r.t. the parameters of the model (uncertainties, priors, costs) can be efficiently computed. In the second part of our method, this allows us to utilize the neural network within a statistical model of an actor's behavior to perform inference about model parameters (Section 3.2).

### 3.1 AMORTIZING BAYESIAN DECISION-MAKING USING NEURAL NETWORKS

Because the Bayesian decision-making problem stated in Eq. (1) is intractable for general cost functions, we approximate it using a neural network $a^* \approx f_\psi(\boldsymbol{\theta}, m)$, which takes the parameters of the Bayesian model $\boldsymbol{\theta}$ and the observed variable $m$ as input and is parameterized by learnable parameters $\psi$. It can then be used as a stand-in for the computation of the optimal action $a^*$ in down-stream applications of the Bayesian actor model, in our case to perform inference about the Bayesian actor's parameters.

#### 3.1.1 UNSUPERVISED TRAINING

We train the neural network in an unsupervised fashion by using the cost function of the decision-making problem as an unsupervised stochastic training objective. After training, the neural network implicitly solves the Bayesian decision-making problem.

Specifically, we use the expected posterior loss

$$\mathcal{L}(\psi) = \mathbb{E}_{p(s \mid m)} \left[ \mathbb{E}_{p(r \mid f_\psi(\boldsymbol{\theta}, m))} \left[ \ell(r, s) \right] \right]$$

as a training objective. Because the inner expectation depends on the parameters of the neural network, w.r.t. which we want to compute the gradient, we need to apply the reparameterization trick (Kingma & Welling, 2014) and instead take the expectation

$$\mathcal{L}(\psi) = \mathbb{E}_{p(s \mid m)}\left[\mathbb{E}_{p(\epsilon)}\left[\ell\left(r, s\right)\right]\right]$$

over a distribution $p(\epsilon)$ that does not depend on $\psi$ where $r = g\left(f_\psi\left(\boldsymbol{\theta}, m\right), \epsilon\right)$ with some appropriate transformation $g$. For example, in the perceptual decision-making model used later (Section 4.1), $p(r \mid a)$ is log-normal with scale parameter $\sigma_r$, so we can sample $\epsilon \sim \mathcal{N}(0, 1)$ and use the reparameterization $g(a, \epsilon) = \exp\left(\log(a) + \epsilon\sigma_r\right)$.

Now, we can easily evaluate the gradient of the objective using a Monte Carlo approximation of the two expectations,

$$\nabla_\psi \mathcal{L}(\psi) \approx \frac{1}{K}\frac{1}{N}\sum_{k=1}^{K}\sum_{n=1}^{N}\nabla_\psi \ell_{\boldsymbol{\theta}}(r_n, s_k),$$

which makes it possible to train the network using any variant of stochastic gradient descent. In other words, the loss function used to train the neural network that approximates the optimal Bayesian decision-maker is simply the loss function of the underlying Bayesian decision-making problem. All that we require is a model in which it is possible to draw samples from the posterior distribution $s_k \sim p(s \mid m)$ and the response distribution $r_n \sim p(r \mid a)$. This allows us to train the network in an unsupervised fashion, i.e. we only need a training data set consisting of parameters $\boldsymbol{\theta}$ and sensory inputs $m$, without the need to solve for the optimal actions beforehand. The procedure is summarized in Algorithm 1. See Section D.1 for the prior distributions used to generate parameters during training. We used the RMSProp optimizer with a learning rate of $10^{-4}$, batch size of 256, and N = M = 128 Monte Carlo samples per evaluation of the stochastic training objective. The networks were trained for 500,000 steps, and we assessed convergence using an evaluation set of analytically or numerically solved optimal actions (see Section D.2).

### 3.1.2 NETWORK ARCHITECTURE

We used a multi-layer perceptron with 4 hidden layers and 16, 64, 16, 8 nodes in the hidden layers, respectively. We used swish activation functions at the hidden layers (Elfwing et al., 2018). As the final layer, we used a linear function with an output $\mathbf{y} \in \mathbb{R}^3$, followed by a non-linearity $a^* = \text{softplus}(y_1 m^{y_2} + y_3)$, where $m$ is the observation received by the subject. This particular non-linearity is motivated by the functional form of the analytical solution of the Bayesian decision-making problem for the quadratic cost function as a function of $m$ and $\boldsymbol{\theta}$ (Section C.1) and serves as an inductive bias (Section D.3).

### 3.2 BAYESIAN INFERENCE OF MODEL PARAMETERS

The graphical model in Fig. 1 B illustrates the generative model of behavior in an experiment. Our goal is to infer $p(\boldsymbol{\theta} \mid \mathcal{D})$, where $\mathcal{D} = \{s_i, r_i : i = 1, \ldots, n\}$ is a dataset of stimuli and responses. We assume that for every stimulus $s$ presented to the subject, they receive a stochastic measurement $m \sim p(m \mid s)$. They then solve the Bayesian decision problem given above, i.e. they decide on an action $a$. We used the neural network $a^* \approx f(m, \boldsymbol{\theta})$ to approximate the optimal action. The chosen action is then corrupted by action variability to yield a response $r \sim p(r \mid a)$. To sample from the researcher's posterior distribution over the subject's model parameters $p(\boldsymbol{\theta} \mid \mathcal{D})$, we use the Hamiltonian Monte Carlo algorithm NUTS (Hoffman et al., 2014). This gradient-based inference algorithm can be used because the neural network is differentiable with respect to the parameters $\boldsymbol{\theta}$ and sensory input $m$. The procedure is summarized in Algorithm 2.

### 3.3 IMPLEMENTATION

The method was implemented in `jax` (Frostig et al., 2018), using the packages `equinox` (Kidger & Garcia, 2021) for neural networks and `numpyro` (Phan et al., 2019) for probabilistic modeling and inference. Our implementation is publicly available at `https://github.com/RothkopfLab/naba`. Our software package enables the user to define new parametric families of cost functions and train neural networks to approximate the decision-making problem and perform Bayesian inference about its parameters. Training a neural network for 500,000 steps took 10 minutes, and drawing 20,000 posterior samples for a typical dataset with 60 trials took 10 seconds.

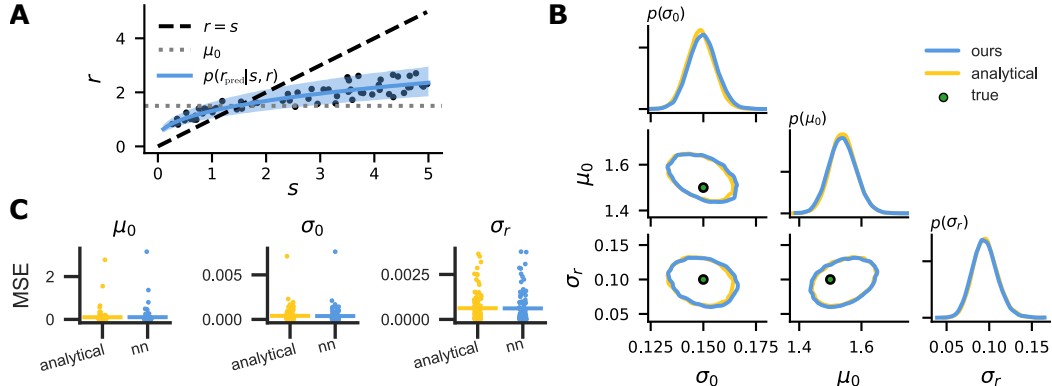

Figure 2: **A** Simulated data from the Bayesian actor model with quadratic cost function are shown as a scatter plot. The posterior predictive distribution $p(r_{\text{pred}} \mid s, r)$ (mean and 94% CI) obtained using our method is shown as a blue line with shaded region. **B** Posterior distributions of parameters obtained using the analytical solution or the neural network to compute optimal actions. The top plot of each column shows the respective marginal posterior distribution for each parameter. Ground truth parameter values are shown for comparison. **C** Evaluation (MSE between posterior mean and ground truth) for multiple runs with uniformly sampled ground truth parameters.

## 4 RESULTS

We evaluate our method on a perceptual decision-making task with log-normal prior, likelihood and action distribution (Section 4.1), which we later combine with several cost functions. For certain cost functions, this Bayesian decision-making problem is analytically solvable. We evaluate our method's posterior distributions against those obtained when using the analytical solution for the optimal action (Section 4.2). Our evaluations allowed us to find possible identifiability problems between prior and cost parameters inherent to Bayesian actor models, which we analyze in more detail (Section 4.3). Finally, we apply our method to real data from a sensorimotor task performed by humans and show that it explains variability and biases in the data (Section 4.5).

### 4.1 PERCEPTUAL DECISION-MAKING MODEL

We now make the decision-making problem more concrete by introducing a log-normal model for the perceptual and action uncertainties. This model is applicable to many different tasks involving perception and action, with a wide range of stimuli, such as time (Yi, 2009; Roberts, 2006), space (Longo & Lourenco, 2007), sound (Sun et al., 2012), numerosity (Roberts, 2006; Longo & Lourenco, 2007; Dehaene, 2003), and different motor actions such as throwing a ball (Willey & Liu, 2018), shooting a hockey puck (Neupärtl et al., 2020), or producing a certain force (Onneweer et al., 2016).

**Sensory measurement** We assume that the observer's sensory measurements are generated from a log-normal distribution $m \sim \text{Lognormal}(m \mid s, \sigma)$. This assumption is motivated by Weber's law, i.e. that the variability scales linearly with the mean (Weber, 1831).

**Posterior distribution** Assuming a log-normal prior $s \sim \text{Lognormal}(\mu_0, \sigma_0)$ and a log-normal likelihood $m \sim \text{Lognormal}(s, \sigma)$, the posterior is $p(s \mid m) = \text{Lognormal}(\mu_{\text{post}}, \sigma_{\text{post}})$, with

$$\sigma_{\text{post}}^2 = \left( \frac{1}{\sigma_0^2} + \frac{1}{\sigma^2} \right)^{-1}, \quad \mu_{\text{post}} = \exp\left( \sigma_{\text{post}}^2 \left( \frac{\ln \mu_0}{\sigma_0^2} + \frac{\ln m}{\sigma^2} \right) \right) \tag{3}$$

This can be shown by using the equations for Gaussian conjugate priors for a Gaussian likelihood in logarithmic space and then converting back to the original space.

**Response distribution** As a probability distribution for the variability of responses $r$ given an intended action $a$, we again use a log-normal distribution $r \sim \text{Lognormal}(r \mid a, \sigma_r)$. This assumption is motivated by the idea of signal-dependent noise in actions (Sutton & Sykes, 1967; Schmidt et al., 1979; Harris & Wolpert, 1998),

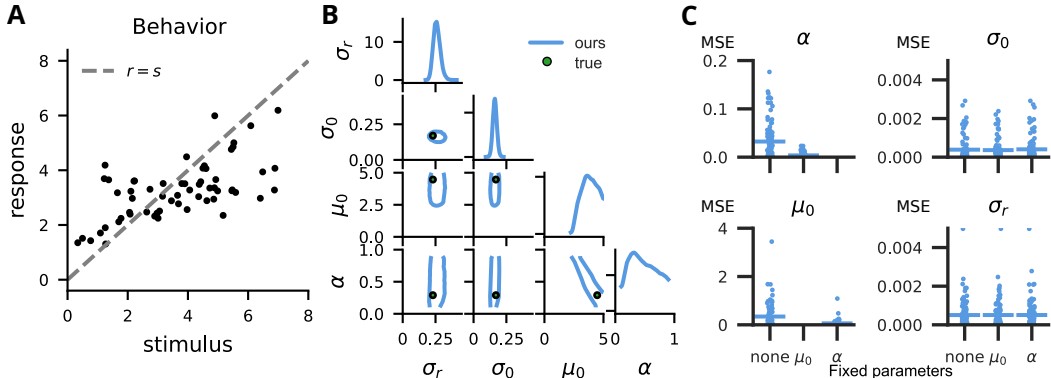

Figure 3: **A** Simulated behavior from the asymmetric quadratic cost (Eq. (4)). The responses exhibit undershots, which could be due to the prior or due to the cost. **B** Posteriors distributions: for each pair of parameters, the plot shows the contours of the 94%-HDI. The pairwise posterior between prior mean $\mu_0$ and cost asymmetry parameter $\alpha$ shows a strong correlation. **C** MSE between ground truth and posterior mean when fixing no parameters, the cost asymmetry parameter $\alpha$ or the prior parameter $\mu_0$. Fixing one of the confounding variables results in an improvement of accuracy in the inference of the other variable. The remaining parameters maintain their accuracy.

### 4.2 EVALUATION USING SYNTHETIC DATA

We first evaluate our method on a case for which we know the analytical solution for the optimal action: the quadratic cost function (see Section C.1 for a derivation). We generated a dataset of 60 pairs of stimuli $s_i$ and responses $r_i$ using the analytical solution for the optimal action with ground truth parameters $\mu_0 = 1.5, \sigma_0 = 0.15, \sigma = 0.2, \sigma_r = 0.1$. The simulated data are shown in Fig. 2 A, with a characteristic pattern of signal-dependent increase in variability, an overshot for low stimulus values and an undershot for higher stimulus values due to the prior. We then computed posterior distributions for the model parameters using the analytical solution for the optimal action and using the neural network. We drew 20,000 samples from the posterior distribution in 4 chains, each with 5,000 warmup steps. Note that, because the concrete values of $\sigma$ and $\sigma_0$ are unidentifiable even when using the analytical solution for the optimal action (see Section F.1), we kept $\sigma$ fixed to its true value during inference. This was done to evaluate our method on a version of the model, for which the analytical solution as a gold standard produces reliable results. Fig. 2 B shows that both versions recover the true parameters, and the contours of both posteriors align well. The posterior predictive distribution generated from the neural network posterior reproduces the pattern of variability and bias in the data (shaded region in Fig. 2 A).

To ensure and quantitatively assess that the method works for a wide range of parameter settings, we then simulated 100 sets of parameters sampled uniformly (see Section D.1 for the choice of prior distributions). For each set of parameters, we simulated a dataset consisting of 60 trials. We then computed posterior distributions for each dataset in two ways: using the analytical solution for the optimal action and using the neural network to approximate the optimal action. In both cases, we drew 20,000 samples (after 5,000 warm-up steps) from the posterior distribution in 4 chains and assessed convergence by checking that the R-hat statistic (Gelman & Rubin, 1992) was below 1.05. The mean squared errors between the posterior mean and the ground truth parameter value in Fig. 2 C show that the inference method using the neural network recovers the ground truth parameters just as well as the analytical version. Fig. F.3 A&B additionally show the error as a function of the ground truth parameter value and Table F.1 shows the results from Fig. 2 C numerically.

### 4.3 LIMITS OF IDENTIFIABILITY OF COSTS AND PRIORS

We can now apply our method to new cost functions, for which analytical solutions are not available. For example, we consider an asymmetric version of quadratic cost (*AsymQC*, see Table E.1) that penalizes overshooting a target more strongly then undershooting it, or vice versa:

$$\ell(r, s) = 2|\alpha - \mathbb{1}(r - s)|(r - s)^2 \quad \text{with} \quad \mathbb{1}(x) = \begin{cases} 1 & \text{if} \quad x \geq 0 \\ 0 & \text{else} \end{cases}. \tag{4}$$

This cost function introduces another source of biases besides perceptual priors: people might undershoot either because of a prior belief that favors closer targets or because of a cost for overshooting (see Fig. 3).

Using our method, we can turn to investigating whether we can tease these different sources of biases apart. Fig. 3 A shows a pattern of behavior with an undershot with ground truth parameters $\mu_0 = 4.47, \sigma_0 = 0.17, \sigma = 0.22, \sigma_r = 0.23$ and $\alpha = 0.29$. Fig. 3 B shows that the ground truth parameters are contained with in the posterior distribution. However, the undershot can either be attributed to a subject trying to avoid overshoots, or to biased perception due to a low prior mean. This is difficult to disentangle, and leads to a correlated posterior for the cost parameter $\alpha$ and the prior mean $\mu_0$ (Fig. 3 B). Over 100 simulated datasets with a range of different ground truth parameter values, the MSE between the inferred posterior mean and the ground truth is higher when both $\mu_0$ and $\alpha$ are unknown. Once we fix one of the confounding parameters at their true value and exclude them from the set of inferred parameters, we observe an increase in accuracy of the inferred parameters (see Fig. 3 C).

Nevertheless, this unidentifiability is a property of the model itself and not a shortcoming of the inference method with the neural network approximation of the optimal action. In fact, our method opens up the possibility to investigate these properties of Bayesian actor models in the first place. To demonstrate this, we repeated this analysis for a quadratic cost function with a quadratic effort cost term (see Section F.2, for which we have derived an analytical solution) and showed that the posteriors obtained using the neural network match those obtained with the analytical solution (see Fig. 3 B).

## 4.4 DISENTANGLING PRIORS AND COSTS

The result in Section 4.3 might sound disappointing. If we do not know the subject's prior or cost, we cannot easily infer them both from a simple experiment. Fortunately, the situation can be remedied by experimental design, as we will now show in simulations. If we introduce different levels of perceptual uncertainty, e.g. by decreasing the contrast of the target, we can disentangle the effects of priors and costs. The intuition is as follows: At different levels of perceptual uncertainty, the actor's prior has a different influence on their posterior. The cost function, however, does not affect the shape of the posterior distribution itself, but only the optimal action under the posterior distribution. Having multiple different levels of prior or cost in an experiment should therefore resolve the unidentifiability, as also proposed by Wei & Hahn (2024).

We simulated an experiment following this intuition. We set the effort parameter of the quadratic cost with quadratic effort to $\beta = 0.9$, and the parameters of the log-normal sensorimotor model to $\mu_0 = 1.5, \sigma_0 = .2$, and $\sigma_r = 0.15$. We chose two levels of perceptual uncertainty $\sigma \in \{0.1, 0.2\}$ and simulated 45 trials for each level (see Fig. 4 A). With this experimental design, both prior mean $\mu_0$ and effort cost $\beta$ can be estimated with good precision (see Fig. 4 B, magenta curves).

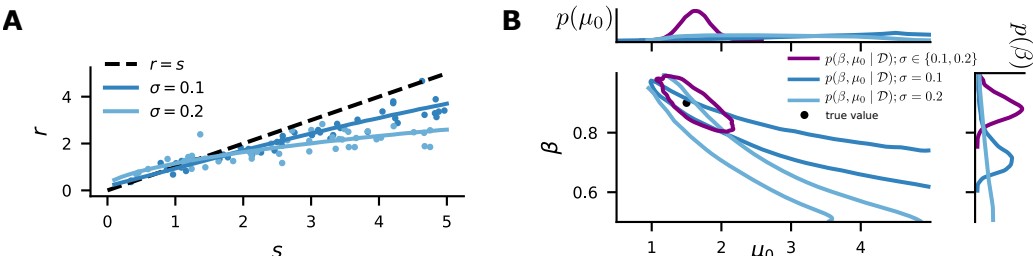

Figure 4: Disentangling priors and costs using different levels of perceptual uncertainty. **A** Data simulated with two different levels of perceptual uncertainty $\sigma$. **B** Inferred posterior distributions (94% CIs) for an experiment with 45 trials of each of two different levels of perceptual uncertainty (purple) and with 90 trials of one level of perceptual uncertainty (shades of blue).

To rule out that just one of the two levels of perceptual uncertainty would have been enough to disentangle prior and cost in this example, we simulated an experiment with an equivalent number of trials in total for each individual level of perceptual uncertainty. Compared to the experiment with two different levels, the posterior distributions showed higher uncertainty in $\mu_0$ and $\hat{\beta}$, and stronger correlation between the two parameters (see Fig. 4 B, blue curves).

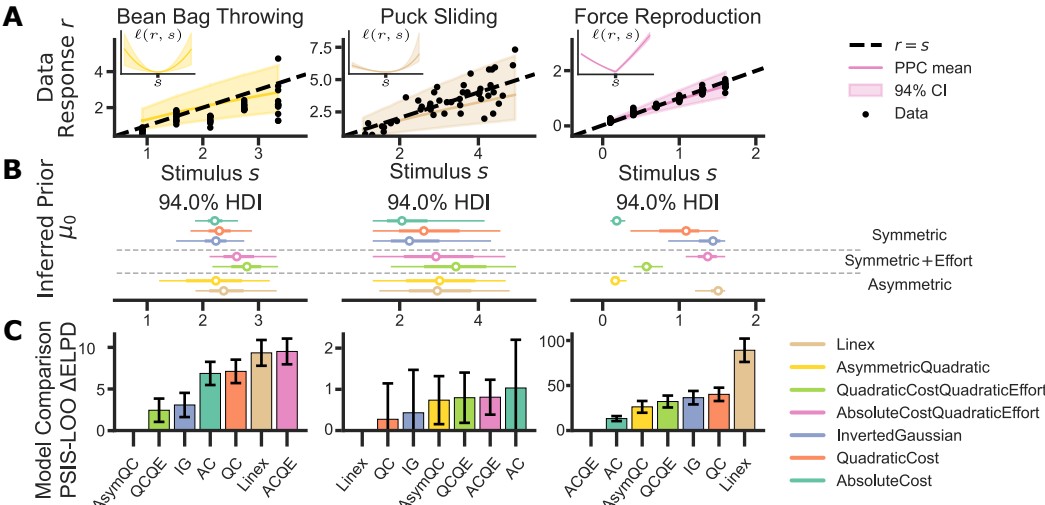

Figure 5: Data of exemplary subjects from three different tasks. The tasks were throwing bean bags at a target (Willey & Liu, 2018), sliding a puck to a target (Neupärtl et al., 2020) and producing a force of certain magnitude (Onneweer et al., 2016), from left to right. Rows show different aspects of the data. **A** Data of the subject with mean and 94% confidence interval of posterior predictive distributions along with a qualitative plot of the cost function with the inferred mean parameters (thick) and the inferred 94%-HDI bounds (light background) over the response $r$ with fixed stimulus $s$ in the upper left panel. **B** Inferred means and 94%-CI for prior mean $\mu_0$ for a range of different cost functions, grouped by their functional form as *symmetric*, *symmetric plus effort term* or *asymmetric*. The cost functions are given explicitly in Table E.1. **C** Model comparison based on the differences in PSIS-LOO estimates of the expected log point-wise predictive density (ELPD) of different cost functions to the best scoring cost function (lower is better). Error bars denote standard error.

## 4.5 INFERENCE OF HUMAN COSTS AND PRIORS

The intended use of our method is to serve as a toolbox for modeling behavior that allows for comparison of different modeling choices in Bayesian actor models. Here, we evaluate seven different parametric cost functions (see Table E.1) on empirical data from three different tasks. The tasks used in our evaluation are a bean bag (BB) throwing task (Willey & Liu, 2018), a puck sliding (PU) task (Neupärtl et al., 2020) and a force reproduction (FOR) task (Onneweer et al., 2016). In the BB task, participants were asked to throw a bean bag at five different target distances from 3 to 11 feet (0.9 to 3.4 meters) with 2 feet increments (0.6 meters). In the PU task, participants needed to press a button for a certain amount of time, which resulted in the distance traveled by a simulated puck on a screen. The virtual distances on screen corresponded to 1.0 to 5.0 meters and were uniformly sampled. The participants received visual feedback by seeing how far the puck travelled onscreen. In the FOR task, participants needed to produce forces on a haptic manipulator from 10 to 160 newtons with 30 newtons increments. They received verbal feedback from the instructor and visual feedback about the applied force.

We fit our model to the data from individual subjects and evaluate it with posterior predictive checks and a model comparison over different cost functions, as shown in Fig. 5. To illustrate the effects of different cost functions on the inferred prior belief, we also visualize the inferred posteriors over $\mu_0$ given different cost functions. For each run, we drew 20,000 posterior samples after 5,000 warm-up steps in four chains. We assessed convergence by checking that the R-hat statistic (Gelman & Rubin, 1992) was below 1.05.

Fig. 5 shows an exemplary subject from each task, overall showing different qualitative patterns of behavior. The subject from the BB task has a tendency to overshoot for near targets and undershoots for far targets. This tendency is explained by the inferred prior means $\mu_0$, which are close to the actual mean of targets in the experiment at 2.15 meters. The model comparison shows a slight advantage for the asymmetric cost function. The subject from the PU task has a slight tendency to undershoot the target. The inferred prior means are very uncertain and the model comparison is inconclusive, indicating that the data from this experimental condition is not sufficient to tell different cost functions apart.

Finally, the subject in the FOR task produces the target force quite accurately. In this data set, different cost functions can lead to greatly differing inferences about the subject's prior mean. The model comparison shows that the Linex cost function provides the worst fit, while the absolute cost with quadratic effort best explains the data. These results illustrate how our method can be used in future work for extensive analyses of different cost functions across a large variety of tasks and subjects.

## 5 DISCUSSION

First, we have presented a new method for Bayesian inference about the parameters of Bayesian actor models. Computing optimal actions in these models is intractable for general cost functions and, therefore, previous work has often focused on cost functions with analytical solutions, or derived custom tools for specific tasks. We propose an unsupervised training scheme for neural networks to approximate Bayesian actor models with general parametric cost functions. The approach is extensible to cost functions with other functional forms suited to particular tasks. Second, performing inference about the parameters of Bayesian actor models given behavioral data is computationally very expensive because each evaluation of the likelihood requires solving the decision-making problem. By plugging in the neural network approximation, we can perform efficient inference. Third, a very large number of tasks involve continuous responses, including economic decision-making ('how much would you wager in a bet?'), psychophysical production ('hit the target'), magnitude reproduction ('reproduce the duration of a tone'), sensorimotor tasks ('reproduce the force you felt'), and cross-modality matching ('adjust a sound to appear as loud as the brightness of this light'). Therefore we see very broad applicability of our method in the behavioral sciences including cognitive science, psychology, neuroscience, sensorimotor control, and behavioral economics. Fourth, over the last few years, a greater appreciation of the necessity for experiments with continuous responses has developed (Yoo et al., 2021). Such decision problems are closer to natural environments than discrete forced-choice decisions, which have historically been the dominant approach. We provide a statistical method for analyzing continuous responses, which allows for quick and exhaustive exploration of the model space spanned by design choices for the Bayesian actor model (e.g. cost functions), and show this on empirical data from three different tasks. Thereby, our method aims to advance scientific progress in our understanding of the internal processes and drivers of human behavior in perceptual decision-making problems. Finally, although the optimality assumption has been common in perceptual and economic decision-making, motor control, and other cognitive tasks, it has been repeatedly disputed (Rahnev & Denison, 2018). In fact, people are not always optimally tuned to the statistics of the task at hand or act only to fulfill the instructed task. This is precisely the motivation for inverse decision-making methods. Instead of postulating optimality, these methods infer under which assumptions the behavior would have been optimal, and thereby conceptually reconcile normative and descriptive models of decision-making while capturing behavior quantitatively.

Once we move beyond quadratic cost functions, identifiability issues between prior and cost parameters can arise. As recognized in other work as well (Acerbi et al., 2014; Sohn & Jazayeri, 2021), these identifiability issues in Bayesian models have implications for how experiments should be designed. We have shown that, when either priors or costs are known, the identifiability issue vanishes. Furthermore, we have shown in simulations that multiple levels of perceptual uncertainty can disentangle the behavioral effects of priors and costs. Based on these results, we recommend experiments with multiple conditions that employ perceptual uncertainties of different magnitudes, between which one can assume either priors or costs to stay fixed, in order to disentangle their effects. Our methodology should prove particularly useful for investigating task configurations that lead to or avoid such unidentifiabilities and can be used to perform inference give data from such experiments.

**Limitations** The strongest limitation we see with the present approach is that we require a model of the perceptual problem that allows drawing samples from the observer's posterior distribution, which works for stimuli that can be described well by these assumptions, e.g. magnitude-like stimuli with log-normal distributions considered in our experiments. Ideally, we would also like to apply the method to perceptual stimuli for which these assumptions do not hold (e.g. circular variables). We see a potential to extend our method by learning an approximate posterior distribution together with the action network. This idea is closely related to loss-calibrated inference (Lacoste-Julien et al., 2011; Kuśmierczyk et al., 2019), an approach that learns variational approximations to posterior distributions adapted to loss functions. We will explore this connection more in future work.

## ACKNOWLEDGMENTS

We thank Nils Neupärtl for initial work on this project idea (Neupärtl & Rothkopf, 2021). We acknowledge the suggestion by an anonymous NeurIPS reviewer to try unsupervised training. We thank Chéla Willey and Zili Liu for sharing their beanbag throwing data (Willey & Liu, 2018), Bram Onneweer for sharing the force reproduction data (Onneweer et al., 2016), and Fabian Tatai for his help in analyzing the puck sliding data (Neupärtl et al., 2020). This research was supported by the European Research Council (ERC; Consolidator Award "ACTOR"-project number ERC-CoG-101045783), by the "The Adaptive Mind", funded by the Excellence Program of the Hessian Ministry of Higher Education, Science, Research and Art, and the Hessian research priority program LOEWE within the project "WhiteBox". We gratefully acknowledge the computing time provided to us on the high-performance computer Lichtenberg at the NHR Centers NHR4CES at TU Darmstadt.

## ETHICS STATEMENT

Methods for inferring beliefs and costs from behavioral data have potential for both positive and negative societal impact. We see a great benefit for behavioral research from methods that provide estimates of people's perceptual, motor, and cognitive properties from continuous action data, particularly in clinical applications where such tasks are easier for patients than classical forced-choice tasks. On the other hand, methods to infer people's uncertainties and costs could potentially be used in harmful ways, especially if they are used without the subjects' consent. The inference methods presented here are applicable to controlled experiments, and are far from scenarios with harmful societal outcomes. Still, applications of these methods should of course take place with the subjects' informed consent and the oversight of ethical review boards.

## REPRODUCIBILITY STATEMENT

We have described our methods in detail in Section 3. Specifically, the training procedure (Section 3.1.1), network architecture (Section 3.1.2), Bayesian inference method (Section 3.2), and implementation (Section 3.3) are described in individual sections. In the appendix, we also provide a pseudo-code version of the algorithms (Section A), detailed graphical models describing the probabilistic models (Section B), and analytical derivations for optimal actions of two cost functions (Section C) . Additional details about the prior distributions used for training the neural networks, and for performing the inference and evaluation are provided in Section D.1 and about the evaluation dataset in Section D.2. We provide an implementation of our methods with instructions for setup and running the code at `https://github.com/RothkopfLab/naba`.

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

## A  ALGORITHM

---

**Algorithm 1** Train neural network to approximate a Bayesian actor model

---

**Output:**  Trained neural network $f_\psi(\boldsymbol{\theta}, m) \approx \arg\min_a \mathbb{E}_{p(s \,|\, m)} \left[ \mathbb{E}_{p(r \,|\, a)} \left[ \ell(r, s) \right] \right]$

**Input:**  Researcher's prior distribution for training the neural network $p(\boldsymbol{\theta})$,
    Subject's perceptual model $p_{\boldsymbol{\theta}}(m \,|\, s)$ and $p_{\boldsymbol{\theta}}(s)$,
    Subject's response model $p_{\boldsymbol{\theta}}(r \,|\, a)$,
    Subject's cost function $\ell_{\boldsymbol{\theta}}(r, s)$
1: **while** test error $>$ desired test error **do**
2:    Draw $\boldsymbol{\theta} \sim p(\boldsymbol{\theta})$
3:    Compute expected loss $\mathcal{L}(\psi) = \mathbb{E}_{p_{\boldsymbol{\theta}}(s \,|\, m)} \left[ \mathbb{E}_{p(\epsilon)} \left[ \ell_{\boldsymbol{\theta}}(r, s) \right] \right] \approx \frac{1}{K} \frac{1}{N} \sum_{k=1}^{K} \sum_{n=1}^{N} \ell_{\boldsymbol{\theta}}(r_n, s_k)$
4:    Perform gradient descent step with $\nabla_\psi \mathcal{L}(\psi) \approx \frac{1}{K} \frac{1}{N} \sum_{k=1}^{K} \sum_{n=1}^{N} \nabla_\psi \ell_{\boldsymbol{\theta}}(r_n, s_k)$
5:    Evaluate error of optimal actions on an evaluation set (see Section D.2)
6: **end while**

---

**Algorithm 2** Bayesian inference about the parameters of a Bayesian actor model

---

**Output:**  Samples from posterior distribution $p(\boldsymbol{\theta} \,|\, \mathcal{D})$

**Input:**  Researcher's prior distribution $p(\boldsymbol{\theta})$,
    Data $\mathcal{D} = \{s_i, r_i : i = 1, \ldots, n\}$,
    Subject's perceptual model $p_{\boldsymbol{\theta}}(m \,|\, s)$ and $p_{\boldsymbol{\theta}}(s)$,
    Subject's response model $p_{\boldsymbol{\theta}}(r \,|\, a)$,
    Subject's cost function $\ell_{\boldsymbol{\theta}}(r, s)$
1: Obtain pre-trained neural network $a = f_\psi(\boldsymbol{\theta}, m)$ from Algorithm 1
2: Sample from the posterior $p(\boldsymbol{\theta} \,|\, \mathcal{D})$ defined by the graphical model Fig. 1 C using NUTS

---

## B  PERCEPTUAL DECISION-MAKING MODEL

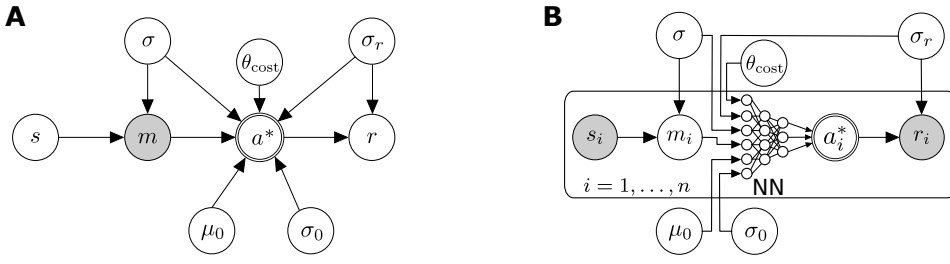

Figure B.1: The graphical models from Fig. 1 with specific parameters for the log-normal perceptual decision-making model described in Section 4.1. The parameters represent the perceptual uncertainty $\sigma$, the motor variability $\sigma_r$ and the log-normal prior defined by $(\mu_0, \sigma_0)$. The parameter vector $\theta_{\text{cost}}$ is defined by the employed cost function. E.g., for quadratic cost without any free parameters, $\theta_{\text{cost}} = \{\varnothing\}$, or for costs function incorporating cost parameters in equations equation 6 and equation 4, it is $\theta_{\text{cost}} = \{\beta\}$ and $\theta_{\text{cost}} = \{\alpha\}$, respectively. **A** Subject view of the task. **B** Model from the researcher's perspective.

## C  DERIVATIONS OF OPTIMAL ACTIONS

### C.1  QUADRATIC COST

One cost function for which the Bayesian decision-making problem under a log-normal observation model and a log-normal response model (Section 4.1) can be solved in closed form is the quadratic

function. In that case, we can write the expected loss as

$$\mathbb{E}\left[\ell(r,s)\right] = \mathbb{E}\left[(s-r)^2\right]$$
$$= \mathbb{E}\left[s^2\right] - 2\mathbb{E}\left[s\right]\mathbb{E}\left[r\right] + \mathbb{E}\left[r^2\right],$$

assuming independence between $s$ and $r$.

Using the moment-generating function of the log-normal distribution, we can evaluate these expectations as

$$\mathbb{E}\left[\ell(r,s)\right] = e^{2\ln\mu_{\text{post}}+2\sigma_{\text{post}}^2} - 2e^{\ln\mu_{\text{post}}+\frac{\sigma_{\text{post}}^2}{2}}e^{\ln a+\frac{\sigma_r^2}{2}} + e^{2\ln a+2\sigma_r^2}.$$

By differentiating with respect to $a$

$$\frac{\partial}{\partial a}\mathbb{E}\left[\ell(r,s)\right] = 2e^{2\sigma_r^2}a - 2\mu_{\text{post}}e^{\frac{1}{2}(\sigma_r^2+\sigma_{\text{post}}^2)}$$

and setting to zero, we obtain the optimal action

$$a^* = \mu_{\text{post}}\,\exp\left(\frac{1}{2}\left(\sigma_{\text{post}}^2 - 3\sigma_r^2\right)\right).$$

Inserting the posterior mean (Eq. (3)), we obtain

$$a^* = \exp\left(\sigma_{\text{post}}^2\left(\frac{\ln\mu_0}{\sigma_0^2} + \frac{\ln m}{\sigma^2}\right)\right)\exp\left(\frac{1}{2}\left(\sigma_{\text{post}}^2 - 3\sigma_r^2\right)\right)$$

$$= \mu_0^{\frac{\sigma_{\text{post}}^2}{\sigma_0^2}}\exp\left(\frac{1}{2}\left(\sigma_{\text{post}}^2 - 3\sigma_r^2\right)\right)m^{\frac{\sigma_{\text{post}}^2}{\sigma^2}} \tag{5}$$

## C.2 Quadratic cost with quadratic action effort

We now consider a class of cost functions of the form

$$\ell(r,s) = \beta(s-r)^2 + (1-\beta)r^2.$$

This allows us to write the expected cost as

$$\mathbb{E}\left[\ell(r,s)\right] = \mathbb{E}\left[\beta(s-r)^2 + (1-\beta)r^2\right]$$
$$= \beta\mathbb{E}\left[(s-r)^2\right] + (1-\beta)\mathbb{E}\left[r^2\right]$$

Using the previously obtained result of Section C.1, we can write this as

$$\mathbb{E}\left[\ell(r,s)\right] = \beta\left(e^{2\ln\mu_{\text{post}}+2\sigma_{\text{post}}^2} - 2e^{\ln\mu_{\text{post}}+\frac{\sigma_{\text{post}}^2}{2}}e^{\ln a+\frac{\sigma_r^2}{2}} + e^{2\ln a+2\sigma_r^2}\right) + (1-\beta)e^{2\ln a+2\sigma_r^2}.$$

Differentiating

$$\frac{\partial}{\partial a}\mathbb{E}\left[\ell(r,s)\right] = 2e^{2\sigma_r^2}a - 2\beta\mu_{\text{post}}e^{\frac{1}{2}(\sigma_r^2+\sigma_{\text{post}}^2)}$$

and setting to zero as well as inserting the posterior mean (Eq. (3)) yields

$$a = \beta\mu_{\text{post}}\exp\left(\frac{1}{2}\left(\sigma_{\text{post}}^2 - 3\sigma_r^2\right)\right)$$

$$= \beta\exp\left(w_0\ln\mu_0 + \frac{\sigma_{\text{post}}^2}{2} - \frac{3\sigma_r^2}{2}\right)m^{w_m}$$

with $w_0 = \frac{\sigma_{\text{post}}^2}{\sigma_0^2}$ and $w_m = \frac{\sigma_{\text{post}}^2}{\sigma^2}$.

# D    HYPERPARAMETERS AND OTHER METHODS DETAILS

## D.1    PARAMETER PRIOR DISTRIBUTIONS

We use relatively wide priors to generate the training data for the neural networks to ensure that they accurately approximate the optimal action over a wide range of possible parameter values:

- $\sigma \sim \mathcal{U}(0.01, 0.5)$
- $\sigma_r \sim \mathcal{U}(0.01, 0.5)$
- $\sigma_0 \sim \mathcal{U}(0.01, 0.5)$
- $\mu_0 \sim \mathcal{U}(0.1, 7.0)$

During inference, we use narrower, but still relatively uninformed prior distributions, to avoid the regions of the parameter space on which the neural network has not been trained:

- $\sigma \sim \text{Half-Normal}(.25)$
- $\sigma_r \sim \text{Half-Normal}(.25)$
- $\sigma_0 \sim \text{Half-Normal}(.25)$
- $\mu_0 \sim \mathcal{U}(0.1, 5.0)$

To evaluate the inference procedure, we use parameters sampled from priors, which correspond to the actual parameter values that we would in expect in a behavioral experiment:

- $\sigma \sim \mathcal{U}(0.1, 0, 25)$
- $\sigma_r \sim \mathcal{U}(0.1, 0, 25)$
- $\sigma_0 \sim \mathcal{U}(0.1, 0, 25)$
- $\mu_0 \sim \mathcal{U}(2.0, 5.0)$

In contrast to the sensorimotor parameters, we kept the same priors for cost parameters during training of the neural network, inference and evaluation:

- Quadratic cost with linear effort: $\beta \sim \mathcal{U}(0.5, 1.0)$
- Quadratic cost with quadratic effort: $\beta \sim \mathcal{U}(0.5, 1.0)$
- Asymmetric quadratic cost: $\alpha \sim \mathcal{U}(0.1, 0.9)$

## D.2    EVALUATION DATASET

To assess convergence of the neural networks, we generated an evaluation dataset consisting of 100,000 parameter sets and optimal solutions of the Bayesian decision-making problem for each cost function. If an analytical solution was available (e.g. quadratic cost), we computed the optimal action analytically. If there was no analytical solution known to us, we solved the Bayesian decision-making problem numerically. Specifically, we computed Monte Carlo approximations of the posterior expected loss

$$\mathcal{L}(a) = \mathbb{E}_{p(s \mid m)}\left[\mathbb{E}_{p(r \mid a))}\left[\ell(r, s)\right]\right]$$

$$\approx \frac{1}{K}\frac{1}{N}\sum_{k=1}^{K}\sum_{n=1}^{N}\ell(r_n, s_k)$$

with $N = K = 10,000$ and used the BFGS optimizer (implemented in `jax.scipy`) to solve for the optimal action $a^*$.

### D.3 INDUCTIVE BIAS FOR THE NEURAL NETWORK

We know the closed-form solution for the optimal action $a^*$ as a function of the parameters $\boldsymbol{\theta}$ for the quadratic cost function (Section C.1). Therefore, we assume that the parametric form of the optimal action as a function of the sensory measurement $m$ will not be substantially different for other cost functions, although the specific dependence on the parameters will vary. We rewrite the optimal action for the quadratic loss (Eq. (5)) as

$$a^* = f_1(\boldsymbol{\theta})m^{f_2(\boldsymbol{\theta})} + f_3(\boldsymbol{\theta}).$$

To allow for additive biases as well, we have added an additive term, which is zero for the quadratic cost function.

## E COST FUNCTIONS USED IN MODEL COMPARISON

The different cost functions in the evaluation on empirical data from Section 4.5 are shown in more detail in Table E.1. Parameters $\gamma$, $\beta$ and $\alpha$ are free parameters in the respective cost functions and are also inferred along with the other latent sensorimotor variables.

| Functional Type | Name | Cost Function $\ell(r, s)$ |
|---|---|---|
| Symmetric | Absolute Cost (AC) | $\|r - s\|$ |
| | Quadratic Cost (QC) | $(r - s)^2$ |
| | Inverted Gaussian (IG) | $1 - \exp\left\{-\frac{(r-s)^2}{2\gamma^2}\right\}$ |
| Symmetric with effort | Absolute Cost Quadratic Effort (ACQE) | $\beta\|r - s\| + (1 - \beta)r^2$ |
| | Quadratic Cost Quadratic Effort (QCQE) | $\beta(r - s)^2 + (1 - \beta)r^2$ |
| Asymmetric | Asymmetric Quadratic (AsymQC) | $2\|\alpha - \mathbb{1}(r - s)\|(r - s)^2$ |
| | Linex | $\frac{2}{\alpha^2}(\exp\{\alpha(r - s)\} - \alpha(r - s) - 1)$ |

Table E.1: Different cost functions used in fitting our model to empirical data in Section 4.5.

The heaviside function $\mathbb{1}(x)$ is given by

$$\mathbb{1}(x) = \begin{cases} 1 & \text{if} \quad x \geq 0 \\ 0 & \text{else} \end{cases}$$

# F  ADDITIONAL RESULTS

## F.1  INFERENCE WITH BOTH PERCEPTUAL AND PRIOR UNCERTAINTY AS FREE PARAMETERS

During evaluation of our method, we found identifiability issues inherent to Bayesian actor models between the prior uncertainty $\sigma_0$ and the sensory variability $\sigma$. In order to produce the same behavior as the one observed, only the ratio $\frac{\sigma_0}{\sigma}$ needs to be inferred correctly (see diagonal correlation in the joint posterior of the two parameters in Fig. F.1 A) – the absolute magnitude of each of the two parameters does not substantially influence the resulting behavior and thus the solution to the true values conditioned on the observed behavior is an underdetermined problem. This intuitively makes sense, since the resulting behavior is largely shaped by how much influence prior and sensory information have, which is weighted by $\sigma_0$ and $\sigma$, respectively.

We were able to considerably improve accuracy in the inference of these two confounding parameters by fixing one of them at their true value (see Fig. F.1 B). Therefore, we decided to keep the perceptual uncertainty $\sigma$ fixed when probing our method since this corresponds to psychophysically measuring $\sigma$ prior to applying our method and fixing it at the measure value.

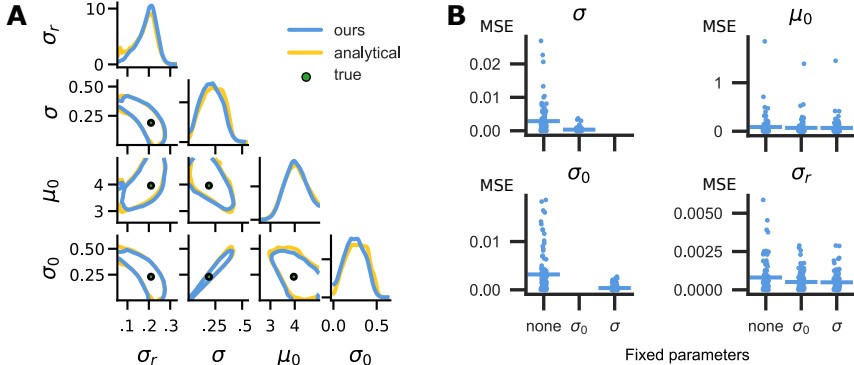

Figure F.1: **A** Posteriors with the analytical model versus the neural network model for an arbitrary, synthetically generated data set that captures the identifiability problem between $\sigma$ and $\sigma_0$ quite clearly. Our method approximates the posterior with the analytical solution quite well and mimics its behavior regarding the uncertainty over different parameters. The pairwise posterior between prior uncertainty $\sigma_0$ and perceptual uncertainty $\sigma$ shows a strong correlation. **B** MSE when fixing no parameters, the perceptual uncertainty $\sigma_0$ or the prior parameter $\sigma$. Fixing one of the confounding variables results in a considerable improvement of accuracy in the inference of the non-fixed parameter. The prior parameter $\mu_0$ maintains its accuracy while the errors for the response variability $\sigma_r$ also slightly decrease.

## F.2 IDENTIFIABILITY OF QUADRATIC COST WITH QUADRATIC ACTION EFFORT

Additionally, we consider a cost function that incorporates the cost of the effort of actions. It is more costly to throw a ball at a longer distance due to the force needed to produce the movement, or it is more effortful to press a button for a longer duration. This can be achieved using a weighted sum of the squared distance to the target stimulus and the square of the response:

$$\ell(r, s) = \beta(s - r)^2 + (1 - \beta)r^2. \tag{6}$$

This cost function also has another source of biases besides perceptual priors: people might undershoot stimuli at larger distances more (see Fig. F.2). An analytical solution of the optimal action for this cost function is derived in Section C.2. Using our method, we can turn to investigating whether we can tease these different sources of biases apart. Fig. F.2 A shows a pattern of behavior with an undershot with ground truth parameters $\mu_0 = 2.95, \sigma_0 = 0.19, \sigma = 0.14, \sigma_r = 0.23, \beta = 0.72$. The posterior distribution Fig. F.2 B shows that the undershot can either be attributed to a subject trying to avoid the mental or physical strain of larger effort, or to biased perception due to a low prior mean. This is difficult to disentangle, and shows in a correlated posterior for the effort cost parameter $\beta$ and the prior mean $\mu_0$. Over 100 simulated datasets with a range of different ground truth parameter values, the MSE between the inferred posterior mean and the ground truth is high when both $\mu_0$ and $\beta$ are unknown. Once we fix one of the confounding parameters at their true value and exclude them from the set of inferred parameters, we observe a considerable increase in accuracy of the inferred parameters (see Fig. F.2 C). Fig. F.3 C&D show the error as a function of the ground truth parameter value and Table F.2 shows the results from Fig. 3 C numerically.

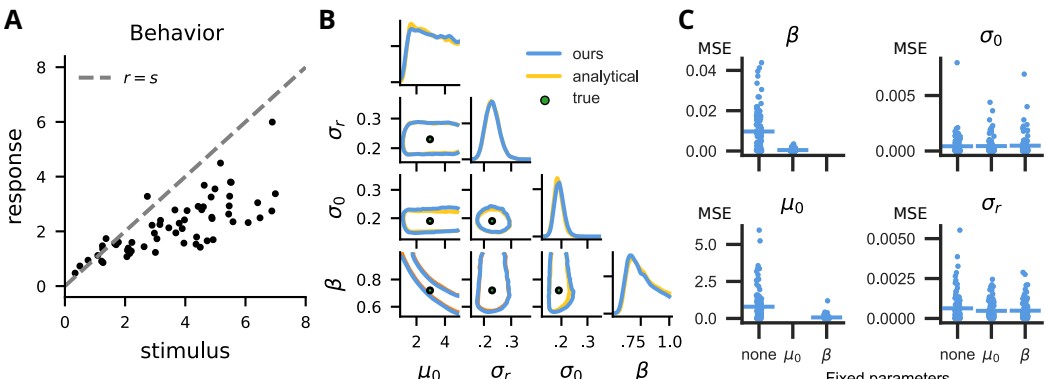

Figure F.2: **A** Simulated behavior from the quadratic cost function with effort cost (Eq. (6)). The responses exhibit undershots, which could be due to the prior or due to the cost. **B** Posteriors with the analytical model versus the neural network model. For each pair of parameters, the plot shows the contours of the 94%-HDI. Our method approximates the posterior with the analytical solution well. The pairwise posterior between prior mean $\mu_0$ and effort cost parameter $\beta$ shows a strong correlation. **C** MSE between ground truth and posterior mean when fixing no parameters, the effort parameter $\beta$ or the prior parameter $\mu_0$. Fixing one of the confounding variables results in an improvement of accuracy in the inference of the other variable. The remaining parameters maintain their accuracy.

## F.3    Accuracy of inference with amortized optimal actions

### F.3.1    Comparison of posterior means and standard deviations

To assess the accuracy of the inference with amortized optimal actions, we used the data sets also shown in Fig. 2 C and Fig. 3 C. We compared means and standard deviations of the posteriors obtained with the analytical solutions for the optimal actions $a^*$ or with the neural network as approximation for the subject's decision-making. The neurally amortized inference accurately produces very similar posteriors to the ones obtained using the analytical solution, as shown in Fig. F.3.

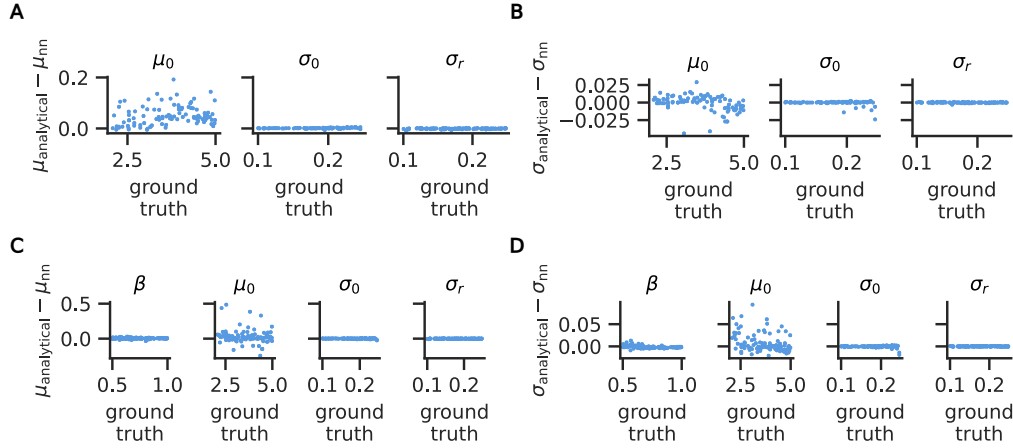

Figure F.3: **Error between posteriors as a function of ground truth parameter values.** For each simulated data set in our evaluation (Fig. 2 C and Fig. 3 C), we computed the difference between posterior means and standard deviations for posterior distributions obtained using analytical optimal actions and our neural network approximations. **A** Difference between posterior means for quadratic cost. **B** Difference between posterior standard deviations for quadratic cost. **C** Difference between posterior means for quadratic cost with quadratic effort. **D** Difference between posterior standard deviations for quadratic cost with quadratic effort.

### F.3.2 MEAN SQUARED ERRORS

We additionally summarize the mean squared errors visualized in Fig. 2 C (Table F.1), Fig. 3 C (Table F.2), and Fig. 3 F (Table F.3).

| Parameter | MSE analytical | MSE NN |
|---|---|---|
| $\mu_0$ | 0.10 | 0.10 |
| $\sigma_0$ | $3.98 \times 10^{-4}$ | $3.77 \times 10^{-4}$ |
| $\sigma_r$ | $6.23 \times 10^{-4}$ | $6.13 \times 10^{-4}$ |

Table F.1: Summary of results for quadratic cost function from Fig. 2 C. Each entry shows the mean squared error (MSE) between the posterior means and ground truth parameters, obtained using the analytical vs. neural network optimal action.

| Parameter | MSE analytical | MSE NN |
|---|---|---|
| $\mu_0$ | 0.74 | 0.77 |
| $\sigma_0$ | $3.63 \times 10^{-4}$ | $4.08 \times 10^{-4}$ |
| $\sigma_r$ | $4.87 \times 10^{-4}$ | $4.86 \times 10^{-4}$ |
| $\beta$ | $8.97 \times 10^{-4}$ | $8.97 \times 10^{-3}$ |

Table F.2: Summary of results for quadratic cost with quadratic effort from 3 C. Each entry shows the mean squared error (MSE) between the posterior means and ground truth parameters, obtained using the analytical vs. neural network optimal action.

| Parameter | MSE (NN) |
|---|---|
| $\mu_0$ | 0.34 |
| $\sigma_0$ | $3.87 \times 10^{-4}$ |
| $\sigma_r$ | $5.05 \times 10^{-4}$ |
| $\alpha$ | $3.23 \times 10^{-2}$ |

Table F.3: Summary of results for asymmetric quadratic cost 3 F. Each entry shows the mean squared error (MSE) between the posterior means and ground truth parameters, obtained using the neural network optimal action.

