# OpenReview forum: "Inverse decision-making using neural amortized Bayesian actors"
_ICLR.cc/2025/Conference — ICLR 2025 Poster_

### Official Review · Reviewer_nJXd · 2024-11-03

**Soundness:** 3
**Presentation:** 3
**Contribution:** 2
**Rating:** 6
**Confidence:** 2

**Summary:**

This paper considers an important and fundamental problem of inferring priors and costs from behavior. Typically, the inverse decision-making problem is intractable. Therefore, the author approximate the solution with a neural network, and show that the ground truth can be recovered well on simulated dataset. The author also explore the human behavioral data.

**Strengths:**

1. The writing is clear, and the core ideas are well articulated.
2. This paper introduces a novel approach for Bayesian inference about the parameters of Bayesian actor models.

**Weaknesses:**

The most significant concern is the lack of experimental advancements. This work only presents experimental results from numerical simulations and some simple human behavioral dataset, where simple MLP is able to recover posterior distributions. Presumably, the algorithm proposed by the author will face several challenge when we have to due with high-dimensional input.
1. It might be hard to train $f_{\psi} (\theta, m)$ when $\theta$ contains more than 100M parameters.
2. The HMC might not have the property of rapid mixing.

**Questions:**

Please refer to the Weaknesses.

---

> ### Author Response · Authors · 2024-11-21
>
> Thank you for taking the time to review our paper from the broader perspective of machine learning in general. We have the feeling that the key idea and concept of our method in the context of computational cognitive science, computational economics, and computational neuroscience has not come across, unfortunately. We will respond to each of your comments and questions below in more detail.
>
>
> **[Lack of empirical advancements]**
>
> We understand the concern that empirical advancements are an important objective, but we want to stress several points here:
>
> 1. First, we would like to direct the reviewer’s attention to the empirical results depicted in Figure 5. These results provide evidence that much previous research on cost functions in sensorimotor control has relied on oversimplifying assumptions, that task-dependent cost functions have much more structure than previously assumed, and that subjective internal effort costs may grow neither linearly nor quadratically.
>
> 2. While we do have results regarding human behavior, see the previous point 1), we are proposing a new method in this paper. We carefully validated the method and applied it to empirical data, see the above point. In our view, the emphasis of our contribution lies in the development of the method and completely novel empirical insights in a scientific domain such as sensorimotor control are beyond the scope of the current paper. However, our method should lead to new empirical insights from experiments in cognitive science and neuroscience.
>
> 3. Moreover, there is empirical advancement from a methodological perspective. By uncovering identifiability issues and how they can be overcome, we provide experimentalists with guidelines on how to design experiments such that they can extract reasonable results from their data (Section 4.3, Figure 4). This is especially important since, e.g., many experiments investigating sensorimotor or economic actions to date do not incorporate different levels of perceptual uncertainty, which we identified as a requirement for identifiability between costs and priors as confounding behavioral parameters.
>
>
> **[Behavioral dataset is too simple]**
>
> We understand the concern that the behavioral experiments to which we applied our method might appear simple, since each trial involves subjects perceiving a single variable and performing a single response. However, the cognitive processes involved are by no means simple. Historically, experimental designs are largely limited to discrete actions, even though natural behavior is continuous. Once we move beyond discrete actions, additional cognitive processes such as action costs and motor variability play a role. There are no established state-of-the-art tools for inferring what drives human behavior in these more natural tasks, and we are confident that our method can contribute to bridging this gap. The relevance of such tasks e.g. in neuroscience has been put forward in recent reviews that have called for new experiments, to which our method should be applicable (Yoo, Hayden, Pearson, 2021).
>
> **[Challenge of high-dimensional models]**
>
> With all due respect, we do not understand where the idea arises that we would want to perform inference over models in the order of 100M parameters. Could the reviewer please clarify?
>
> We clearly state in the paper that we are concerned with models from cognitive science, neuroscience, and behavioral economics. Models in this domain typically have few parameters, usually within the range of 2–12 parameters. These models gain their strength not by the number of parameters, but instead by the interpretability of the parameters as perceptual, cognitive, or motor factors. Importantly, it is desirable to be able to manipulate experimental variables to test and verify the influences of parameters on behavior. All related studies that we mention in Section 2 as well as all examples referenced in the discussion or used in our evaluations only have a few parameters.
>
> More concretely, Bayesian models of perception often have a few parameters exactly like the model we consider in our evaluations: perceptual uncertainty, prior beliefs, behavioral costs, and motor variability. To give a few examples, some models have two free parameters (Petzschner & Glasauer, 2011; Neupärtl et al., 2021). Even more complex Bayesian models of perception only have up to ten (Battaglia et al., 2011) or twelve (Acerbi et al., 2014) free parameters.

---

> ### Author Response · Authors · 2024-11-21
>
> **[HMC might not have the property of rapid mixing]**
>
> This is a very general concern about HMC, which, in our opinion, is misdirected since such general criticism could be applied to any study that uses this method. The main focus of our paper is on using neural networks to amortize Bayesian decision-making problems. This enables the use of any gradient-based sampler for the inverse decision-making problem. In particular, our method does not require the use of HMC, but can make use of any state-of-the-art sampler that uses the log posterior probability (and its gradient).
>
> Nevertheless, we understand concerns about MCMC convergence. To check the convergence of our Markov chains, we verified that the $\hat{r}$ statistic was below 1.05 in all cases.
>
> Additional references:
> - Yoo, S. B. M., Hayden, B. Y., & Pearson, J. M. (2021). Continuous decisions. Philosophical Transactions of the Royal Society B, 376(1819), 20190664.
> - Petzschner, F. H., & Glasauer, S. (2011). Iterative Bayesian Estimation as an Explanation for Range and Regression Effects: A Study on Human Path Integration. The Journal of Neuroscience, 31(47), 17220–17229.
> - Battaglia, P. W., Kersten, D., & Schrater, P. R. (2011). How Haptic Size Sensations Improve Distance Perception. PLoS Computational Biology, 7(6), e1002080.
> - Neupärtl, N., Tatai, F., & Rothkopf, C. A. (2021). Naturalistic embodied interactions elicit intuitive physical behaviour in accordance with Newtonian physics. Cognitive Neuropsychology, 38(7–8), 440–454.
> - Acerbi, L., Vijayakumar, S., & Wolpert, D. M. (2014). On the origins of suboptimality in human probabilistic inference. PLoS computational biology, 10(6), e1003661.]

---

> > ### Comment · Reviewer_nJXd · 2024-11-27
> >
> > Thank you for your thoughtful response. While I am not yet fully convinced that the methodology will consistently perform well on more advanced human behavior tasks, I consider this a promising direction for future research. I have decided to raise my score in light of this.

---

> > > ### Author Response · Authors · 2024-11-27
> > >
> > > We would like to thank you very much for your updated assessment in response to our rebuttal. Please let us know if there are any additional points that you would like us to clarify.

---

### Official Review · Reviewer_QV14 · 2024-11-04

**Soundness:** 3
**Presentation:** 4
**Contribution:** 4
**Rating:** 6
**Confidence:** 2

**Summary:**

The paper addresses the challenge of using Bayesian models to infer decision-making parameters (inverse decision making) from behavioral data, especially for tasks involving continuous actions where traditional Bayesian methods struggle with computational intractability.

The authors propose a new method where a pre-trained neural network, trained unsupervisely, was used to approximate an actor model’s parameter. The gradient-based Bayesian inference makes the method relatively efficient. This approach shows promising alignment with analytical solutions where they exist and effectively models human behavioral data in various sensorimotor tasks.

**Strengths:**

The paper provides an innovative approach by using a neural network to approximate the Bayesian model for inverse inference, which traditionally faces computational intractability issues. Their neural network method, trained in an unsupervised manner, enables efficient inference of decision-making parameters without relying on closed-form solutions or restrictive assumptions (like Gaussian distributions or quadratic costs).

Clear problem formulation and motivation.

**Weaknesses:**

The authors mentioned that their method could be applicable to a large number of tasks involving continuous responses, including economic decision-making, psychophysical production and crossmodality matching. However, the authors only tested their method on sensorimotor tasks. Testing methods on a diverse set of tasks involving continuous responses would significantly strengthen the paper.

The authors acknowledge that this method is currently constrained to relatively straightforward perceptual models. Extending it to more complex tasks (such as those involving circular variables or advanced cognitive reasoning) remains a limitation in its current form.

**Questions:**

How scalable is the current approach? What are the computational requirements for training the neural networks for more complex cognitive reasoning tasks?

Could the authors provide more details about the choice of network architecture and hyperparameters?

---

> ### Author Response · Authors · 2024-11-21
>
> Thank you for taking the time to review our paper. We appreciate your feedback and will go through it one by one in the following.
>
>
> **[Extension to different tasks and overall applicability]**
>
> We understand your concerns about the applicability of our method to other domains. However, even if the method were only applicable to sensorimotor tasks, this would, in our view, be a major contribution. The reason is that ample research has investigated human sensorimotor control using a few canonical cost functions (e.g. Kording & Wolpert, 2004) and here we extend the cost functions that can be considered considerably allowing for parameterized effort costs, which is e.g. implicitly common in resource-rational approaches. A recent (Sohn & Jazayeri, 2021) publication also noted the difficulty of such inferences, for which the present manuscript proposes a resolution. But, beyond this broad applicability, we are confident in our model’s generalizability, because the formal setting’s only constraint is that the agent has uncertainty over a latent variable and performs an action associated with a cost. This is also the case, for example, in behavioral economics, where models with “cognitive uncertainty” have seen a recent surge in popularity (e.g. Khaw et al., 2021; Enke & Graeber, 2023; Barretto-Garcia et al., 2023). In these models, the log-normal assumption we use to model the likelihood of the sensory measurement $m$ has also become increasingly common (Khaw et al., 2021; Enke & Graeber, 2023). This assumption will also generalize to other domains and modalities where Fechner’s Law holds, e.g. numerosity (internal cognitive representations), distance and size perception (visual), pitch perception (auditory), etc.
>
> In cases where the log-normal assumption does not hold, e.g. circular variables, the log-normal assumption can be dropped in the perceptual component and our method could still be used with any other distribution, as long as there is a closed form of the subject’s posterior belief about the stimulus $p(s | m)$. For circular variables, this could be the Von Mises distribution, for which conjugate priors (Guttorp & Lockhart, 1988) exist. Our method could then be used to perform model comparison over different perceptual distributions and cost functions. Of course, we are also interested in cases where there is no analytical expression for the posterior distribution, and we have addressed this as future work in the discussion section.
>
> It would be helpful if you could clarify what you mean by “advanced cognitive reasoning tasks”. We would be happy to discuss relevant models and how they relate to the method we are proposing.
>
>
> Additional references:
> Guttorp, P., & Lockhart, R. A. (1988). Finding the location of a signal: A Bayesian analysis. Journal of the American Statistical Association, 83(402), 322-330.
>
>
> **[Scalability]**
>
> This is an important point, thank you for the question. The amount of training required will likely scale with the number of parameters. While the scalability of the network is an empirical question, we are confident that the training should not be too computationally expensive with more parameters. Currently, training of a network takes less than 30 minutes and converges quickly on the CPU of a standard laptop computer (e.g. with Intel Core i7-8565U CPU). The network architecture is rather small and could easily be extended to deeper and more expressive networks. Thus, in terms of architecture and training time, we get excellent results while still being multiple orders of magnitude below the computational cost of most modern machine learning applications.
>
> For the kinds of models we have in mind, the number of parameters should not increase too much. Additional parameters could be introduced in the cost function, but the number of parameters should remain rather low to guarantee explainability. For more complex stimuli (e.g. multidimensional stimuli like color spaces), perceptual priors and cost functions with as many parameters as there are dimensions could be possible. If there are concrete “more complex cognitive reasoning tasks” you have in mind, we would be happy to discuss those.

---

> ### Author Response · Authors · 2024-11-21
>
> **[Clarification of network architecture and hyperparameters]**
>
> This is all provided in the paper. Section 3.1.1, 3.1.2 and 3.3 specify the training procedure, architecture and implementation details, respectively. Appendix A provides pseudocode for training the neural network, Appendix D.1 specifies the distributions for generating training data and Appendix D.3 explains an inductive bias used in the output layer.
>
> In summary, we use four hidden layers with [16, 64, 16, 8] nodes and swish activation functions. We train the network in an unsupervised fashion directly on the cost function by sampling random parameter configurations for 500,000 steps. We use the RMSProp optimizer with a learning rate of $10^{-4}$ and a batch size of 256.
>
> Our reasoning for these choices was that they were sufficient for good performance on the models we investigated in this paper. Of course, future work could investigate different architectures and training regimes in terms of data efficiency or computational cost. One could even think about network architecture search, but this is beyond the scope of the current paper.

---

### Official Review · Reviewer_bwX5 · 2024-11-04

**Soundness:** 2
**Presentation:** 2
**Contribution:** 3
**Rating:** 6
**Confidence:** 3

**Summary:**

This paper introduces a method for performing Bayesian inference on the parameters of Bayesian observer-actor models, particularly suited for scenarios where Bayesian decision-making can be computationally intractable. The approach leverages a neural network to amortize the decision-making process of the subject by training the network to minimize the expected task-relevant cost with respect to the posterior over latent states and the action distribution. This setup allows for efficient, gradient-based inference of parameters from behavioral data. The authors validate their approach on synthetic data, highlighting its effectiveness and also discuss identifiability issues with recommendations to mitigate them. They further illustrate the method's applicability to human behavioral data.

**Strengths:**

This paper address an important bottleneck in inverse decision-making by amortizing the agent's behavior using a neural network. This enables efficient Bayesian inference over the subject's behavioral model parameters. The experiments on synthetic data validate the approach through comparison with analytical solutions. The discussion on identifiability and the experiment design recommendations add valuable practical insights.

**Weaknesses:**

In this work, the proposed approach aims to infer what a subject’s decisions were optimal for. However, there is still an assumption of optimal behavior, which may not always hold in real-world scenarios. Factors such as suboptimal learning or changing task demands can lead to deviations from optimality. Even if these deviations could potentially be reframed as an alternative optimality criterion, doing so would introduce additional identifiability challenges. It would be beneficial  to discuss the limitations of this assumption.

Another potential limitation is that the use of the reparameterization trick requires a specific form of action distribution, which may restrict the model’s adaptability to diverse datasets and tasks where this distributional form does not apply.

Finally, the presentation could be improved. Several figures lack clear labels and legends, making them difficult to interpret, and acronyms are introduced without prior definition. A revised presentation with attention to these details would enhance the paper.

**Questions:**

- In figure 2A, could you clarify what is $r^{\ast}$? Should it be $a^{\ast}$ instead?

- The top left panel in figure 2B is missing a label. Should it be $\sigma_0$?

- In figure 2C, it would be useful to include separate x-axis labels for the analytical and nn cases.

- The caption for figure 3B uses $\beta$ as cost asymmetry parameter, but all figure labels use $\alpha$. Are they the same?

- In figures 3B and 3C, it would be helpful to make the ranges of the axes the same in all panels.

- Figure 4b is difficult to follow, including a legend would be very helpful.

---

> ### Author Response · Authors · 2024-11-21
>
> Thank you very much for your detailed suggestions on how our work can be improved. We appreciate a lot and will respond to each point separately below.
>
>
> **[Optimality Assumption]**
>
> We respectfully would like to disagree with this point. Yes, we agree that the optimality assumption in forward modeling, i.e., in models from the perspective of the researcher, such as ideal observer, ideal actor, and ideal learner, is a very common one in perceptual decision-making, economic decision-making, motor control, and other cognitive tasks. We also fully agree that people are not always optimally tuned to the statistics of the task at hand, or act to only fulfill the instructed task. However, this is precisely the reason why we have developed methods for inverse decision-making to infer under which assumptions the behavior would have been optimal! Without revealing our identity, we have contributed to inverse modeling over the last twenty years precisely to address “real-world scenarios”. Using such methods, we could show that, indeed, human participants can have false beliefs and idiosyncratic subjective costs, which lead to systematic deviations from “optimal” behavior. Thus, the “suboptimality” can be explained in the inverse modeling framework. Therefore, optimality is not a weakness, but allows inferring the subjective parameters of behavior. We agree that the formulation of behavior as being rational needs careful modeling of the possible influences of individual subjects’ behavior, but inverse models in general, and the one we have developed here, make it possible to investigate the reasons for the deviations from “optimality”.
> We take this comment by the reviewer as motivation to extend the discussion of this point in the revised version of the manuscript and will mention relevant literature on the optimality versus suboptimality debate, e.g. in perceptual decision-making (Rahnev & Denison, 2018).
>
>
> **[Reparameterization Trick]**
>
> For all models with continuous responses we are aware of, and for all related work cited in the paper, exponential family or other distributions amenable to the reparameterization trick should work. In our view, this covers an enormous amount of behavioral experiments in psychology, cognitive science, and neuroscience, including motor control. Thus, the developed methods should be exceedingly useful to the scientific community. If there are relevant models we have missed, it would be great if you could please name examples from the literature. In such cases, recently developed methods for computing gradients of more general probabilistic programs without the reparameterization trick will be useful to compute the gradient of the expected loss (e.g. Lew et al. 2023; https://dl.acm.org/doi/abs/10.1145/3571198).
>
>
> **[Overall presentation & figures]**
>
> Thank you for raising the questions about our figures and labels below. We respond to them one by one and will revise our paper accordingly. We have thoroughly revised the figures also with the points raised below.
>
>
> *[[Figure 2A - Posterior Predictive]]*
>
> Sorry about the confusion in the legend of Fig. 2A. The shaded area is a posterior predictive distribution over the responses performed by the subject, given their actually observed responses. We will rename it as $p(r_{pred} \mid s, r)$ in the revised version.
>
>
> *[[Figure 2B - Missing label]]*
>
> Yes, sorry for omitting the label! This is the marginal distribution of $\sigma_0$. The y-axis will be labeled $p(\sigma_0)$. We will adjust all labels of marginal distributions accordingly.
>
>
> *[[Figure 2C - x-axis labels]]*
>
> Thank you, we provided separate labels on the x-axis for the analytical and the NN model, but recognize that this is hard to see. We will separate the two labels “nn” and “analytical” more clearly to avoid confusion.
>
>
> *[[Figure 3B - Asymmetry Parameter]]*
>
> Yes, thank you for catching this! And sorry, it should indeed have been $\alpha$ also in the caption.
>
>
> *[[Figures 3B, 3C, 4]]*
>
> Thank you. These are good suggestions, which we will use in the revised version.

---

> > ### Comment · Reviewer_bwX5 · 2024-11-27
> >
> > Thank you to the authors for their detailed responses to my comments and questions. I am happy to raise my score with the expectation that the presentation issues will be addressed in the revised version of the paper.

---

> > > ### Author Response · Authors · 2024-11-27
> > >
> > > We would like to sincerely thank you for your updated assessment in response to our rebuttal. In the revised version of the paper, which we have now uploaded, we have tried to address all concrete points raised in your review.
> > >
> > > In particular, we have
> > > - extended the discussion of the optimality assumption and mentioned the relevant literature (l. 515 - 520)
> > > - revised our figures according to the suggestions
> > >     - renamed the posterior predictive in Fig. 2A
> > >     - added labels to the y-axes of the marginal distributions in Fig. 2B
> > >     - separated the labels for “nn” and “analytical” in Fig. 2C
> > >     - fixed the naming of the asymmetry parameter $\alpha$ in the caption of Fig. 3B
> > >     - used the same axis limits for $\sigma_r$ and $\sigma_0$ in Fig. 3B and C (the other two parameters $\mu_0$ and $\alpha$ still have different axis limits because they live on a different scale)
> > >     - added a legend to Fig. 4B to indicate that the curves represent the posterior distributions $p(\beta, \mu_0 | \mathcal{D})$ for different levels of perceptual uncertainty $\sigma$.
> > >
> > > Please let us know if there are any additional points that you would like us to clarify.

---

### Author Response · Authors · 2024-11-21
**Global Response to Reviews**

We would like to thank all reviewers for their reviews. In light of some of the questions and comments, we would like to clarify the conceptual framework of our method: we are not just solving the decision-making problem faced by an agent in a task, but instead, we are taking a so-called doubly-Bayesian approach. This means that we frame both the perceptual problem of a participant in an experiment and the statistical problem of a researcher analyzing the observed behavior of a participant from a Bayesian perspective. In other words, we perform Bayesian inference on the free parameters of Bayesian decision-making models.

Second, the researcher performs Bayesian inference about the parameters of the subject’s internal model and cost function. Thus, our paper deals with the much more difficult problem of inference over the decision-making problem faced by an agent. This is comparable to the difference between solving the reinforcement learning problem versus solving the inverse reinforcement learning problem in sequential decision-making.

Third, inverse modeling does not imply that subjects are optimal from the perspective of the researcher who is analyzing behavioral data. Instead, behavior can be highly suboptimal from the perspective of the researcher and could even be fully random, and this would still be expressible through internal uncertainties and cost functions. Put differently, subjects are always optimal with respect to a certain cost function. However, the cost function a researcher assumes can be vastly different from the cost function a participant implicitly employs, and this is further distorted by perceptual and motor noise and potentially false beliefs. This is the fundamental premise of inverse modeling, which, therefore, fourth, fundamentally reconciles normative (“optimal versus suboptimal”) and descriptive (“data fitting”) models.

Fifth, without revealing our identity, we have contributed over the last 20 years to quite a number of algorithms for inverse modeling precisely to address “real-world scenarios”, i.e., finding the internal beliefs and uncertainties and internal costs and benefits in naturalistic behavior! To pick one specific example, computing Bayes-optimal actions is intractable for more general, structured cost functions, as they are relevant in naturalistic settings. Previous work has often focused on quadratic costs, which admit analytical solutions for certain posteriors, or derived custom tools tailored to a specific task. Here, we propose a new unsupervised training scheme for neural networks to approximate Bayesian actor models with general parametric cost functions, which broadens the range of possible cost functions, e.g., to include subjective internal costs, allowing us to identify resource-rational decisions.

Sixth, performing inference about the parameters of Bayesian actor models given behavioral data is computationally very expensive because each evaluation of the likelihood requires solving the Bayesian actor’s decision-making problem. By utilizing the neural network approximation, we can perform efficient inference in the inner inference loop. Finally, a very large number of tasks involving continuous responses across fields ranging from economic decision-making (‘how much would you wager in the following bet?’) to psychophysical production (‘hit the target’), magnitude reproduction tasks (‘reproduce the duration of the tone you just heard with a button press’) and sensorimotor tasks (‘reproduce the force you just felt with the robotic manipulandum’) or even cross-modality matching tasks (‘adjust a sound to appear as loud as the brightness of this light’).

Therefore, we see very broad applicability of our method in the behavioral sciences including cognitive science, psychology, neuroscience, and behavioral economics. We will update the final version of the paper by more clearly stating the conceptual framework and methodology.
Therefore, in our view, this paper represents a significant step beyond the state of the art in the modeling of decision-making in the fields of computational cognitive science, computational neuroscience, and economics.

We would like to ask the reviewers to additionally see the individual responses we provide below. If our responses can clarify some of the questions posed in the reviews and lead the reviewers to appreciate the value of our method, we would very much appreciate it if the reviewers consider increasing the score.

---

### Meta-Review · Area_Chair_cdqU · 2024-12-26

**Metareview:**

This paper investigate a new method for inverse decision-making in sensorimotor tasks with continuous actions.
The proposed approach aims to infer what are a subject’s optimal decisions, with the assumption of optimal human behavior.
As mentioned by reviewers, the parameterization trick requires a specific form of action distribution, which will restrict the model’s adaptability to diverse online optimization tasks.
The AC is very familiar with Bayesian optimization theory and algorithms, but not psychology. It might be a good paper for a psychology conference instead of an AI paper.

**Additional Comments On Reviewer Discussion:**

After rebuttal, the reviewers consider it marginally acceptable.

---

### Decision · Program_Chairs · 2025-01-22

Accept (Poster)